# Integrating Multiple Models Using Image-as-Documents Approach for Recognizing Fine-Grained Home Contexts [note 1]

**DOI:** 10.3390/s20030666

**Published:** 2020-01-25

**Authors:** Sinan Chen, Sachio Saiki, Masahide Nakamura

**Affiliations:** 1Graduate School of System Informatics, Kobe University, 1-1 Rokkodai-cho, Nada, Kobe 657-8501, Japan; sachio@carp.kobe-u.ac.jp (S.S.); masa-n@cs.kobe-u.ac.jp (M.N.); 2RIKEN Center for Advanced Intelligence Project, 1-4-1 Nihonbashi, Chuo-ku, Tokyo 103-0027, Japan

**Keywords:** context recognition, image, cognitive APIs, machine learning, majority voting, score voting, range voting, smart home

## Abstract

To implement fine-grained context recognition that is accurate and affordable for general households, we present a novel technique that integrates multiple image-based cognitive APIs and light-weight machine learning. Our key idea is to regard every image as a document by exploiting “tags” derived by multiple APIs. The aim of this paper is to compare API-based models’ performance and improve the recognition accuracy by preserving the affordability for general households. We present a novel method for further improving the recognition accuracy based on multiple cognitive APIs and four modules, fork integration, majority voting, score voting, and range voting.

## 1. Introduction

As Internet of Things (IoT) and Artificial Intelligence (AI) technologies continue to develop, people have increasing expectations about smart home services. Recognizing fine-grained contexts within individual houses is a key technology for next-generation smart home services. We use the term “fine-grained” home context to represent a home context that is more concrete and is specifically defined by individual houses, residents, and the environment for special purposes of application, such as elderly monitoring [1,2,3], autonomous security [4], and personalized healthcare [5,6]. It has been studied for many years in the field of ubiquitous computing [7,8]. Traditional ubiquitous computing employs ambient sensors [9,10], wearable sensors [11], and indoor positioning systems [12] that are installed at home to retrieve various data.

In recent years, the emerging deep learning [13,14,15,16] allows the system to recognize multimedia. Since image, voice, and text usually contain richer information than conventional sensor data, it is promising to use such multimedia data for recognizing fine-grained home contexts. In our study, we especially focus on image data with human activities at home for recognising home contexts. For this, one may try to recognize home contexts via image recognition based on naive deep learning. However, constructing a custom recognition model dedicated to a single house requires a huge amount of labeled datasets and computing resources [17,18]. It is not only hard to construct a universal recognition model from one house to another, but also the security and privacy issues that come with a large amount of data influence acceptability for the users. Thus, there is still a big gap between research and real life.

Our interest is to use little image data, applying cognitive services to implement affordable context sensing that can adapt to custom contexts in every single house. The cognitive service is a cloud service with cognitive computing functions that provide the capability to understand multimedia data (i.e., vision (object recognition) [19], speech recognition [20], natural language processing [21], etc.), based on sophisticated machine-learning algorithms powered by the offered big data and large-scale computing. They are offered by cloud companies, but the data processing algorithms behind them are not public. They are also widely used in various fields of research, such as modern knowledge management solutions [22] and criminal detection and recognition [23]. The cognitive API is an application program interface via the HTTP/REST protocol [24], with which developers can easily integrate powerful recognition features in their own applications. An image-based cognitive API receives an image from an external application, i.e., extracts specific information from the image, and returns the information in JavaScript Object Notation (JSON) [25] format from the cloud server rather than the local. The information usually contains a set of words called “tags”, representing objects and concepts that the API has recognized in the given image. Examples of tags from the API are: [Living, room, indoors, classroom, basement, supporting structure]. The information of interest and the way of recognizing the image vary among individual cognitive services. Related work uses image tagging technology with deep learning, as in [26,27,28], but the implementation is more complex. In our future realistic implementation, for security and privacy of the users the images are not saved after sending the APIs over.

The main contribution of this paper is to present a novel method which is not only affordable but also has higher accuracy in recognizing fine-grained home contexts. For this purpose, we are currently investigating techniques that integrate inexpensive camera devices, multiple image-based cognitive APIs, and light-weight machine learning. We previously encoded the tags of a single API to document vectors, then applied them into machine learning for the model construction [29]. However, we found that the accuracy significantly decreased for contexts with multiple people (e.g., “General meeting”, “Dining together”, “Play games”). In this paper, we define a concept called “image as documents”, which uses different cognitive APIs for receiving the same image. As the proposed method, we present four modules, fork integration and three voting approaches (i.e., majority [30], score [31], range [32]), to integrate multiple models generated from different APIs. In this way, we can not only compare the difference in API-based models’ performance, but improve the recognition accuracy. Furthermore, we also discuss the potential implementation of the model simplification as in [33], of the more efficient process as in [34], and of the other techniques as in [35] and [36], integrated into the proposed method.

In order to evaluate the proposed method, we experimented to recognize the seven contexts of our laboratory, which use little image data and five cognitive APIs. Based on the proposed method, we completed the process from each independent model construction to multiple model integration, and implemented the above four modules. As a result, for each API-based model, the Imagga API-based model performed best within the five models, and the ParallelDots API-based model was the worst. Meanwhile, the overall accuracy by majority voting reached 0.9753. Furthermore, the overall accuracy by score voting reached 0.9776. We also checked the accuracy distribution by range voting, which was meant to solve the problem of recognition instability in contexts with multiple people. In this way, we found that the top of overall accuracy reached 0.9833 when the value of the lower limit was between 0.5 and 0.6. Thus, the recognition accuracy was significantly improved by the method proposed in our experiment.

## 2. Related Work

The problem of recognizing fine-grained home contexts with human activities [37,38,39,40] has been widely studied in the field of ubiquitous computing. As described in the introduction, it is defined by every user depending on a special purpose. However, for realizing a technique that applies for general households, we consider that it should have several advantages at least: (1) Low cost of devices and systems in both purchase and maintenance, (2) light-weight and a high-accuracy approach for data processing, (3) a stable and secure approach for data communication. In this section, we introduce some related works from recent years around the above three points.

Nakamura et al. [41] proposed a system that recognizes the activities of residents using big data accumulated within a smart house. Ueda et al. [42] also proposed an activity recognition system using ultrasonic sensors and indoor positioning systems within a smart house. While the performance of these systems is great, they are still too expensive for general households. Sevrin et al. [43] contributed to the effort of creating an indoor positioning system based on low cost depth cameras (Kinect). However, for retrieving more specific information on human activities at home (e.g., cleaning, dining, etc.), embracing the position of users is obviously not enough. In recent years, activity recognition with deep learning has become a hot topic. Research in [44] built on the idea of 2D representation of an action video sequence by combining the image sequences into a single image called the Binary Motion Image (BMI) to perform human activity recognition. Asadi-Aghbolaghi et al. [45] presented a survey on current deep learning methodologies for action and gesture recognition in image sequences. While deep learning is a powerful approach for recognizing image data, a huge amount of data is required to build a high-quality model. Therefore, it is unrealistic for individual households to prepare a huge amount of labeled datasets for custom fine-grained contexts.

Using cloud services to recognize human activities at home is key for implementing light-weight data processing. Pham et al. [46] presented a Cloud-Based Smart Home Environment (CoSHE) for home healthcare. While the effect of the system is good, various basic sensors and devices must be installed, which is not ideal for implementation and long-term maintenance. Menicatti et al. [47] proposed a framework that recognizes indoor scenes and daily activities using a cloud-based computer vision. Their concept and aim are similar to our method. However, the way of encoding tags is based on a the naive Bayes model where each word is present or not. Moreover, the method is supposed to be executed on a mobile robot, where the image is dynamically changed. Thus, the method and the premise are different from ours. Research in [48] investigates the influence of a person’s cultural information towards vision-based activity recognition at home. The accuracy of the fine-grained context recognition would be improved by taking such personal information into machine learning. We would like to investigate this perspective in future work.

Regarding the security of uploading images to cloud services, Qin et al. [49] studied the design targets and technical challenges that lie in constructing a cloud-based privacy-preserving image-processing system. There are also some the related works that focus on the security and privacy of smart homes. Dorri et al. [50] proposed a blockchain-based smart home framework with respect to the fundamental security goals of confidentiality, integrity, and availability. Geneiatakis et al. [51] employed a smart home IoT architecture to identify possible security and privacy issues for users. Through the above articles, we also plan, in future work, to focus more on making computation and communication practical for the encrypted data of smart homes.

## 3. Methodology

This section produces a complete description on the preliminary study, proposed method, and discussion of the related techniques.

### 3.1. Preliminary Study

Constructing a single classifier model is a basic and essential part of realizing fine-grained home context recognition. Unlike naive deep learning, we previously dedicated the features of images extracted from a single cognitive API, to apply to light-weight supervised machine learning [29]. The key step for building the context recognition model is to make every image document covert for a set of numerical values, which is document vectorization processing (see step 4). In this section, we describe the method that constructs a recognition model from a single API. It also can be used for performance comparison among the different cognitive APIs, which has been developed from our other preliminary study using unsupervised learning [52].

The procedure consists of the following five steps. 


**Step 1: Acquiring data**


A user of the proposed method first defines a set $C=\{c_{1},c_{2},\ldots ,c_{l}\}$ of home contexts to be recognized. Then, the user deploys a camera device in the target space to observe. The user configures the device so as to take a snapshot of the space periodically with an appropriate interval.


**Step 2: Creating datasets**


For each context $c_{i}\in C$, the user manually selects representative *n* images $IMG\left(c_{i}\right)=\left\{img_{i1},img_{i2},\ldots ,img_{in}\right\}$ that effectively expose $c_{i}$ from all images obtained in step 1. At this time, the total l×n images are sampled as datasets. Then, the *n* images in $IMG(c_{i})$ are split into two sets, $train(c_{i})$ an $test(c_{i})$, which are the training dataset with α images and the test dataset with n−α images, respectively.


**Step 3: Extracting tags as features**


For every image $img_{ij}$ in $train(c_{i})$, the method sends $img_{ij}$ to an image recognition API, and obtains a set $xTag\left(img_{ij}\right)=\left\{t_{1},t_{2},\ldots \right\}$, where $t_{1},t_{2},\ldots$ are tags that the API has extracted from $img_{ij}$. The method performs the same process for $test(c_{i})$ and obtains $yTag(img_{i^{\prime }j^{\prime }})$. At this step, there is a total of l×n tags in the set.


**Step 4: Converting tags into vectors**


Regarding every $xTag(img_{ij})$ as a document, and the whole tag set as a document corpus, the method transforms $xTag(img_{ij})$ into a vector representation $xVec\left(img_{ij}\right)=\left[v_{1},v_{2},\ldots \right]$, where $v_{r}$ represents a numerical value characterizing the *r*-th tag. Famous document vectorization techniques include TF-IDF [53], Word2Vec [54], and Doc2Vec [55]. The selection of the vector representation is up to the user. Similarly, the method converts $yTag(img_{i^{\prime }j^{\prime }})$ into $yVec(img_{i^{\prime }j^{\prime }})$ using the same vector representation.


**Step 5: Constructing a classifier**


Taking $xVec(img_{ij})$(1≤i≤l,1≤j≤α) as predictors and $c_{i}(1\leq i\leq l)$ as a target label, the method executes a supervised machine learning algorithm to generate a multi-class classifier CLS. For a given vector $v=[v_{1},v_{2},\ldots ]$, if CLS returns a context $c_{i}$, which means that the context of the original image of *v* is recognized as $c_{i}$. The accuracy of CLS can be evaluated by $yVec(img_{i^{\prime }j^{\prime }})$ to see if CLS returns the correct context $c_{i^{\prime }}$.

### 3.2. Proposed Method

To improve recognition accuracy (especially in the case of Figure 1) based on Section 3.1, this section describes the most important question in this paper, which is how to construct a whole recognition model by integrating multiple recognition models (see Figure 2). As we mentioned in Section 1, the core of our method is to use the image-as-documents concept, which operates with a fork integration module and a choice among three voting modules. For ensuring that many results of multiple cognitive APIs accurately correspond to every image, the preliminary stage of the training classifier is also very important.

The specific whole steps are as follows. 


**Step 6: Constructing multiple classifiers**


By repeating steps 3 to 5 of Section 3.1 for different image recognition APIs, the proposed method constructs *m* independent recognition models. Note that training and test datasets created in steps 1 and 2 can be reused and shared among different models. As a result of the model construction, we have a set of classifiers $CLS_{1},CLS_{2},\ldots ,CLS_{m}$.


**Step 7: Add vectorizer for new images**


For each $CLS_{q}$, the method generates a vectorizer $VEC_{q}$, which transforms a given image img into a vector representation xVec(img) through the *q*-th cognitive API. Now, if we input any new image of the target space, the concatenation $VEC_{q}+CLS_{q}$ outputs $c_{i}$ as a predicted context class.


**Step 8: Integrate multiple models**


To complete the model construction, the method first adds a fork integration module *F*, which sends a given image simultaneously to *m* recognition models $VEC_{q}+CLS_{q}(1\leq q\leq m)$, corresponding to the image-as-documents concept. Then, the method adds three voting modules, which users choose in different home contexts, as follows.
**Majority voting**: It receives *m* outputs $c^{1},c^{2},\ldots ,c^{m}$ from $VEC_{q}+CLS_{q}(1\leq q\leq m)$, and returns $mode(c^{1},c^{2},\ldots ,c^{m})$ (see Figure 3).**Score voting**: It receives *m* outputs $c^{1},c^{2},\ldots ,c^{m}$ and scores $s^{1},s^{2},s^{3},\ldots ,s^{m}$ of each output from $VEC_{q}+CLS_{q}$(1≤q≤m), and returns $\max\sum_{i=1}^{m}s^{i}\left(c^{i}\right)$ by comparing the total of scores with the same output $c^{i}$(see Figure 4).**Range voting**: It receives *m* outputs $c^{1},c^{2},\ldots ,c^{m}$ and scores $s^{1},s^{2},s^{3},\ldots ,s^{m}$ of each output from $VEC_{q}+CLS_{q}(1\leq q\leq m)$, and sets a lower limit for scores $s^{i}$ to be used. The output $c^{i}$ will be used in the score voting if the corresponding score $s^{i}$ is above the lower limit (compare Figure 4 and Figure 5).

### 3.3. Discussion

To construct a model using multiple cognitive APIs, another method is combining the tags of different cognitive API models, which come from each image (see Figure 6). However, this will increase the dimension of the input data of the built model. The related techniques for dimensionality reduction include Principal Component Analysis (PCA) [33], Locally Linear Embedding (LLE) [56], Latent Semantic Analysis (LSA) [57] and so on. We would like to experiment with them for model simplification and accuracy in our future work. In addition, for dimension reduction [58,59,60] of document vectors, the Restricted Boltzmann Machines (RBMs) [61,62] is a good method. In the existing research, many models only use a small number of features as input; hence, there may not be enough information to classify documents accurately. Conversely, if more features are input, as we discussed earlier, it will increase the dimension of input data, resulting in a large increase in the training time of the model, and its recognition accuracy may also lower. Therefore, we also would like to use RBMs to extract highly distinguishable features from the combined input features, and use them as input to the corresponding model in our future work. We believe that this will greatly improve the efficiency of model construction. Furthermore, conducting the proposed method using the other machine learning techniques (e.g., Hidden Markov Model (HMM) [63], regression, graphical models [64], etc.) is also feasible. We will conduct experiments to compare these technologies combined with the proposed method in future work.

## 4. Experimental Evaluation

This section introduces results, discussion, and an experiment conducted for comparing the difference in API-based models’ performance and improving the recognition accuracy.

### 4.1. Experimental Setup

The experiment was conducted in a shared space of our laboratory. First, we installed a USB camera in a fixed position to acquire images of the space. We then developed a program that takes a snapshot with the camera every five seconds, and uploads the image to a server. The image resolution is 1280 × 1024. The images were accumulated from July 2018 to March 2019. The target shared space is used by members of our laboratory for various activities. In this experiment, we chose the seven kinds of fine-grained contexts: “Dining together”, “General meeting”, “Nobody”, “One-to-one meeting”, “Personal study”, “Play games”, and “Room cleaning”. The detail of each context is as in Table 1. For each context, we selected and labeled 100 representative images from the server, taken on different dates. We then randomized the order of a total of 700 image data, and split them into half, as training data and test data.

### 4.2. Building and Combining API-Based Models

We first built a recognition model for the seven contexts. The following five cognitive APIs were used to extract tags from images referring to each context: Microsoft Azure Computer Vision API [65], IBM Watson Visual Recognition API [66], Clarifai API [67], Imagga REST API [68], and ParallelDots API [69]. Table 2 and Table 3 show the representative images of seven contexts (including tag results) and USB camera. We can see that the different APIs recognized the same image from different perspectives. Each tag set extracted from an image was then transformed into a vector representation using TF-IDF (Term Frequency–Inverse Document Frequency) [53]. We then imported the datasets and the corresponding context labels to Microsoft Azure Machine Learning Studio [70]. For each cognitive API, we trained a classifier using the Multiclass Neural Network with the default setting. Each of the five trained models was evaluated by the test data to observe the performance of individual models. We finally used four modules to build a whole recognition model by integrating five individual models. To check accuracy distribution, we adjusted the lower limit of scored class probabilities to between 0 and 0.9.

### 4.3. Results

Table 4 shows the accuracy results of each cognitive API-based model and three voting modules. Among the five models, the Imagga API-based model was the best (0.9429), while the ParalleDots API-based model scored the lowest (0.7718). These results can be used as reference values for model performance comparison. As for the context-wise accuracy, the performances of the five models were each different (see Table 4). For instance, let us compare the Watson API-based model and the ParalleDots API-based model. The Watson model was bad at recognizing “General meeting” (0.6730), compared to the ParalleDots model (0.8910). Interestingly, however, the Watson model was better at recognizing “One-to-one meeting” (0.8040) than the ParalleDots model (0.4460).

With regard to the overall accuracy with three voting modules, the majority voting achieved an accuracy of 0.9753, the score voting achieved an accuracy of 0.9776, and the range voting achieved the top accuracy of 0.9833 with the range 0.5 to 0.6. Regarding the context-wise accuracy with the majority voting, these limitations of the individual models were mutually complemented. The recognition accuracy of “Dining together” was 0.9565, “Personal study” was 0.9561, and “Room cleaning” was 0.9572, while the accuracy of “General meeting”, “Nobody”, “One-to-one meeting”, and “Play games” were 1.0000. Regarding the context-wise accuracy with score voting, it further made up for the shortage of simply obtaining the final result by the quantity. The recognition accuracy of “General meeting” was 0.9685, “Personal study” was 0.9751, and “Room cleaning” was 0.9720, while the accuracy of “Dining together”, “Nobody”, “One-to-one meeting”, and “Play games” were 1.0000. Regarding the context-wise accuracy with the range voting, it excludes some low-score API results before voting, which further promotes the improvement of the accuracy. The recognition accuracy of “General meeting” was 0.9836 and of “Personal study” was 0.9800, while the accuracy of “Dining together”, “Nobody”, “One-to-one meeting”, “Play games”, and “Room cleaning” were 1.0000. Figure 7 presents the distribution of accuracy results using range voting within the range 0 to 0.9, and includes the overall accuracy and the context-wise accuracy. We can see the top of the overall accuracy was 0.9833 when the lower limit was between 0.5 and 0.6. The entire accuracy of “Dining together” stabilized at 1.0000. However, the accuracy of “Play games” and “General meeting” were unstable, especially for “Play games”, which had the lowest accuracy, 0.9608.

### 4.4. Discussion

In the proposed method, the recognition accuracy heavily depends on the quality of tags extracted by the cognitive API. The reason why the ParalleDots-based model was bad at “One-to-one meeting” (0.4460) was that (1) no distinctive word characterizing of the context was found, and (2) the number of words in the tag sets was relatively small. The accuracy also depends on the nature of the context. We found that contexts where people are dynamically moving (e.g., “Dining together”, “Room cleaning”) were relatively difficult to recognize. In such contexts, observable features are frequently changed from one image to another; for instance, positions of people, visible furniture and background. Therefore, the API may produce variable tag sets for the same context, which decreases the internal cohesion of the feature vectors.

Including majority voting was a great solution to improve the accuracy. In the typical ensemble learning, the individual classifiers should be weak to avoid overfitting. This is because the classifiers use the same features for the training. in our case, we extract different features by different APIs. Since the individual models are trained by different features, it does not cause the overfitting problem. It was seen from the results of naive majority voting that the accuracy of “Dining together”, “Personal study”, and “Room cleaning” were not perfect. The reason is that some situations of the majority results of an image were wrong. The recognition accuracy of “Personal study” and “Room cleaning” improved significantly using score voting. However, there was greater instability in the contexts with multiple people (e.g., “Dining together”, “General meeting”) compared with the results of majority voting. On adjusting the lower limit of scored class probabilities to between 0 and 0.9, there was instability in the accuracy of “Play games” and “General meeting” but not for “Dining together”. One of the reasons for this is that the context richness of “Dining together” was prominent compared to others. This means the output tags of “Dining together” were many, whether by the total or the semantic (see Table 2). The other reason is there were some difficulties in recognizing the contexts with no big change in the number of persons and objects (e.g., “Play games”, “General meeting”). With regard to the top of the overall accuracy, 0.9833, by range voting with the range 0.5 to 0.6, it means that some situations in the majority results of an image were wrong when the scored class probabilities were less than 0.5 or above 0.6.

## 5. Conclusions

In this paper, a method that integrated models based on multiple cognitive APIs and four presented modules for improving the recognition accuracy is proposed. From experimental evaluation, the difference in API-based models’ performance is compared, confirming the advantage that the recognition accuracy is improved by the proposed method.

The image recognition method is different from one API to another. By constructing multiple classifiers with different perspectives, taking majority voting derives the context with a maximum likelihood for the same image. However, there are also some images with recognition difficulty; hence, the case of the false results being output by the majority APIs. Using score voting, we could reduce false results determined by only the number of outputs within the same context to some extent. Furthermore, by setting the different range of lower limits, we deeply understood the recognition difficulty for each context by finding the range with the highest accuracy. This is of great significance for improving the proposed method in our future study. As to the topic of fine-grained home context recognition for general households, this paper has some points that need to be improved. While the different cognitive APIs were used for building models and performance evaluation, the different experimental spaces have not yet been used. Moreover, to build the model of context recognition in future different households, how to select the representative data of context more scientifically is still a task directly related to recognition accuracy. These are the directions of our future work. In addition, to achieve effective application, it is necessary to consider both the retrieval of more features from the data, and the development of various algorithms. Therefore, investigating more ways to use cloud resources to retrieve feature values of local images will also be a topic of future work.

## Figures and Tables

**Figure 1 sensors-20-00666-f001:**
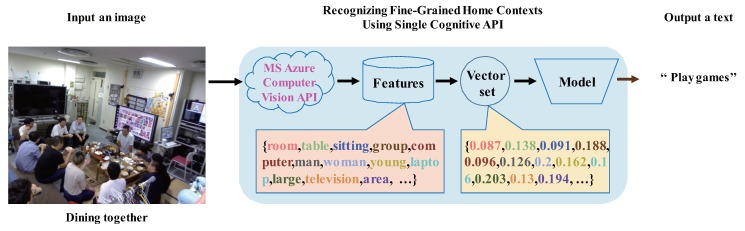
Example of an image with multiple people misrecognized using a single cognitive API.

**Figure 2 sensors-20-00666-f002:**
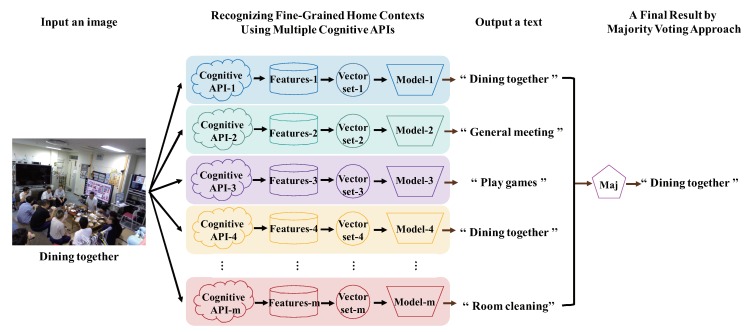
Example of recognizing an image using multiple cognitive APIs and majority voting.

**Figure 3 sensors-20-00666-f003:**
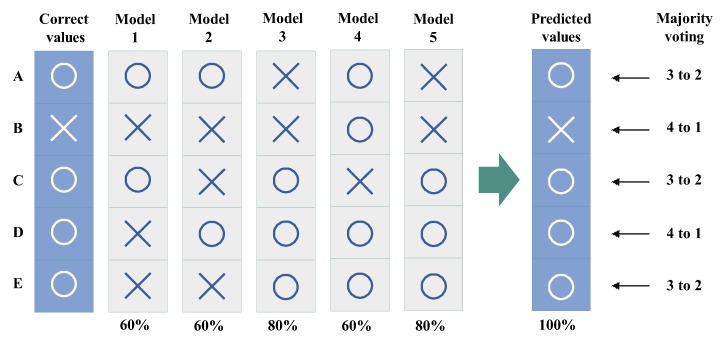
Example of majority voting with ensemble learning.

**Figure 4 sensors-20-00666-f004:**
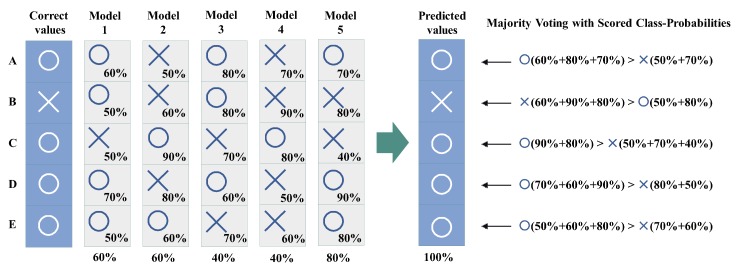
Example of score voting using the total of each class probability.

**Figure 5 sensors-20-00666-f005:**
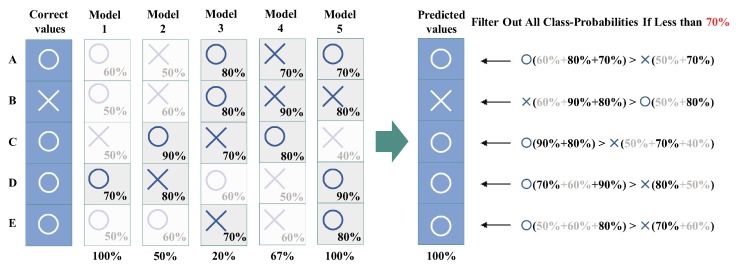
Example of range voting using all class probabilities that scored above 70% based on Figure 4.

**Figure 6 sensors-20-00666-f006:**
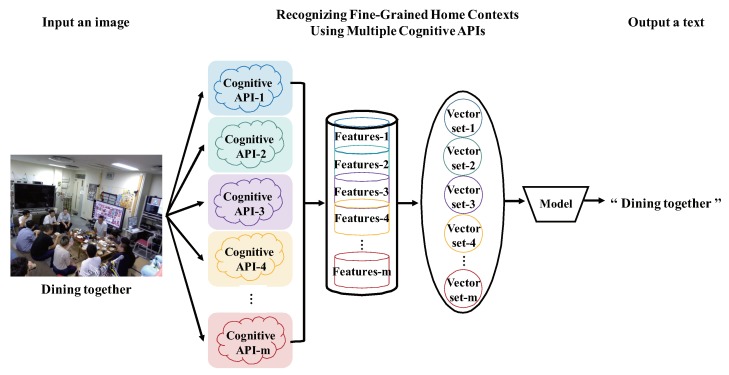
Example of constructing a model by combining the features of multiple cognitive APIs.

**Figure 7 sensors-20-00666-f007:**
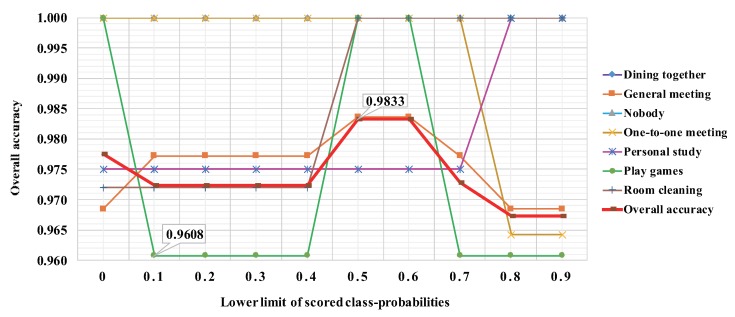
The distribution of accuracy results using range voting within the range 0 to 0.9.

**Table 1 sensors-20-00666-t001:** The detail of each defined context in this experiment.

Context Labels	The Contents of What the Images of Each Context Represent
Dining together	We often cook by ourselves to **dining together** in our laboratory
General meeting	We are sitting together in a **general meeting** every Monday
Nobody	There is also the **nobody** situation during the weekend or holidays
One-to-one meeting	We often have a **one-to-one meeting** for the study discussion
Personal study	Sometimes the public computer is used for **personal study**
Play games	We often gather around and **play games** to relax in our spare time
Room cleaning	The staff twice a week come for **room cleaning** in our laboratory

**Table 2 sensors-20-00666-t002:**
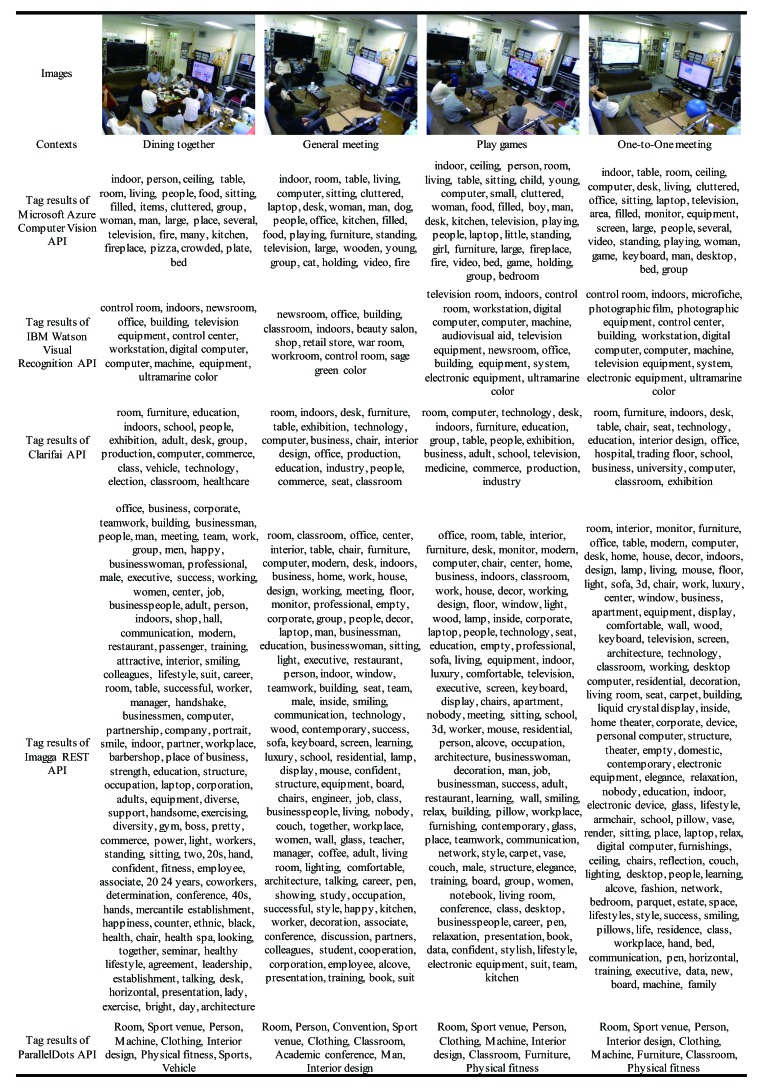
The representative images of the four contexts (including tag results from different APIs).

**Table 3 sensors-20-00666-t003:**
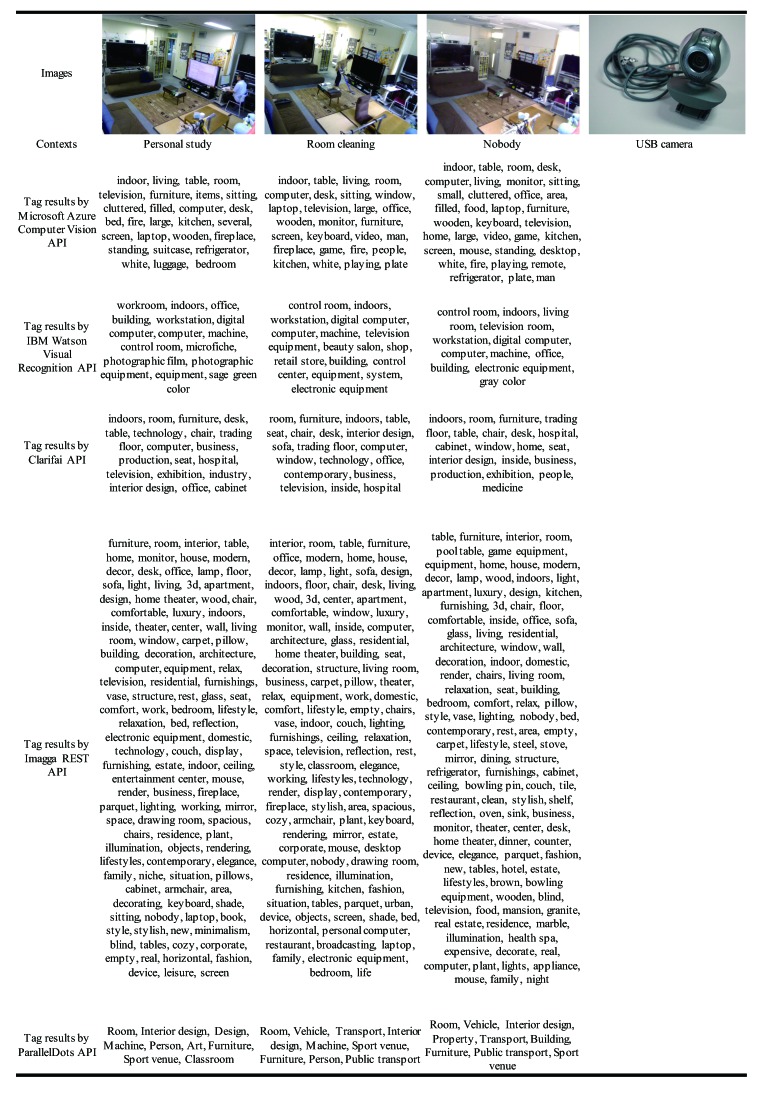
The representative images of the other three contexts (including tag results) and USB camera.

**Table 4 sensors-20-00666-t004:** The accuracy results of each cognitive API-based model and three voting modules.

Model or Voting Names	Overall Accuracy	Dining Together	General Meeting	Nobody	One-to-One Meeting	Personal Study	Play Games	Room Cleaning
Azure API – model	0.8543	0.9550	0.8910	1.0000	0.6610	0.9170	0.8430	0.7650
Watson API – model	0.8000	0.8860	0.6730	0.8230	0.8040	0.9380	0.8040	0.7060
Clarifai API – model	0.9143	0.9090	0.9820	0.9110	0.8390	0.9170	0.9220	0.9220
Imagga API – model	**0.9429**	0.9550	0.9270	1.0000	0.8930	0.9580	0.9220	0.9610
ParalleDots API – model	**0.7718**	0.7950	0.8910	0.9330	**0.4460**	0.8750	0.6670	0.8040
Majority voting	0.9753	0.9565	1.0000	1.0000	1.0000	0.9561	1.0000	0.9572
Score voting	0.9776	1.0000	0.9685	1.0000	1.0000	0.9751	1.0000	0.9720
Range voting (0.5 to 0.6)	**0.9833**	1.0000	0.9836	1.0000	1.0000	0.9800	1.0000	1.0000

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
