# Peer review of "Integrating Multiple Models Using Image-as-Documents Approach for Recognizing Fine-Grained Home Contexts"

_sensors, 2020, doi:10.3390/s20030666_

Round 1

Reviewer 1 Report

This paper present a technique that integrates multiple image-based cognitive APIs 3 and light-weight machine learning. Their key idea is to regard every image as documents by exploiting 4 “tags” derived by multiple APIs and propose a road that use multiple cognitive APIs for ensemble learning.  This paper provide very complete, comprehensive experiments although I did not see much creative model. 

The paper has some minus typesetting problem such as ".." in the middel of 58 rows. 

Reviewer 2 Report

This paper addresses a very interesting research topic. There is no doubt that the potential of image analysis for the characterization of different environments compared to other methods such as those based on sensor data, is a topic of current interest. In this case, the authors focus on home contexts. However, the structure of the paper does not help to understand the interest of the research topic and neither does it help to understand the particular contribution of the research presented by the authors. In order to improve the manuscript the following suggestions are proposed:

Complete Section 1. Introduction with a broader explanation of the problem approach (including sufficient references) that supports the objectives of the work, the interest of the research topic and its relevance for the benefit of society. The objectives of the work should be explained more clearly and in relation to the problem approach. In addition, concepts such as long-term objectives used by the authors should be clarified... What then are the short and medium-term objectives of the work presented? Section 2. Preliminaries does not provide enough information to constitute a separate section. It should be considered as a unique section together with the Introduction. On its own, this section provides very little content. Both Sections 1 and 2 should be rewritten and the manuscript should be structured as follows: (1) Well-argued Introduction; (2) State of the Art (which would contain section 6); (3) Methodology (which would contain sections 3 and 4); (4) Experimental Evaluation; (5) Conclusions Another important issue is to explain more clearly what part of the research presented corresponds to previous work by the authors. This clarification can be made in the explanation of the proposed methodology. The Methodology Section should contain sections 3 (which seems to refer to the authors' previous research and therefore is not the contribution of this article) and 4 (which seems to be the new contribution). This issue should be clarified. In general, the paper is too fragmented. The proposed subsections are excessive. There is practically a section to introduce each definition, which makes it difficult to have an overall view and a logical, developed and complete understanding of the manuscript. The structure of the paper does not help in an easy, global and comprehensive reading of the work, nor of its contribution, nor of the importance of the problem being addressed. As for the proposed method, the authors argue that if majority voting is taken, the context is obtained with the maximum likelihood. However, it is not clear whether the features that make one API more reliable than another are analysed and whether this aspect is considered. In the Experimental Evaluation section, a lot of previously presented information is repeated. It is recommended to remove this repeated information (or remove it from the introduction) and focus more on the results. Finally, it is suggested that the conclusions be rewritten. The conclusions are not a summary of the article and it is hoped that the authors will actually highlight the main conclusions that arise from the research. The references are insufficient. The images are repetitive. Their content does not justify the number of images.

Reviewer 3 Report

This is an interesting paper on fine-grained context recognition. This is an interesting topic with a number of potential applications. The paper is well presented. The authors provide a lot of information in the document and a researcher can learn the basics without spending uncountable time for conducting a lit review. Below I am providing some comments. Thus I would recommend the authors to revise their paper by including the missing information.

Comment #1 

Section 1 is too large. I would recommend shortening this section and move content to Section 2 which is the related work. (The Introduction section is about introducing the concept of the paper and not about the prior work. Please revise accordingly.)

Comment #2

Your current Section 2 should become Section 3, I would suggest that Section 3.1 should discuss prior papers and Section 3.2 should present the algorithms you describe at the current Section 2. 

Comment #3

I like the the "Preliminaries" section. However, I would like to read more information about basic algorithms. Some pseudocode or some sort of table that summarizes things would be quite helpful.

Comment #4

It might make sense to include a subjection with techniques related to "model simplification" that reduce the dimension of the input data I would like to comment that there are other techniques for dimensionality reduction such as PCA, LLE, etc. Did you experiment with them? 

Moreover, it is known that Restricted Boltzmann Machines (RBMs) can be used to preprocess the data and basically to help the "machine learning" process become more efficient. I would suggest the authors discuss in the paper about such a possibility as well as to consider a future implementation. 

Papers that should be added and discussed: 

-- Tenenbaum, J. B., De Silva, V., & Langford, J. C. (2000). A global geometric framework for nonlinear dimensionality reduction. science, 290(5500), 2319-2323. 

-- Cao, L. J., Chua, K. S., Chong, W. K., Lee, H. P., & Gu, Q. M. (2003). A comparison of PCA, KPCA and ICA for dimensionality reduction in support vector machine. Neurocomputing, 55(1-2), 321-336.

-- Roweis, S. T., & Saul, L. K. (2000). Nonlinear dimensionality reduction by locally linear embedding. science, 290(5500), 2323-2326.

-- Belkin, M., & Niyogi, P. (2003). Laplacian eigenmaps for dimensionality reduction and data representation. Neural computation, 15(6), 1373-1396.

-- Ngiam, J., Khosla, A., Kim, M., Nam, J., Lee, H., & Ng, A. Y. (2011). Multimodal deep learning. In Proceedings of the 28th international conference on machine learning (ICML-11) (pp. 689-696). 

-- Mousas, C., & Anagnostopoulos, C. N. (2017). Learning Motion Features for Example-Based Finger Motion Estimation for Virtual Characters. 3D Research, 8(3), 25. 

-- Nam, J., Herrera, J., Slaney, M., & Smith, J. O. (2012, October). Learning Sparse Feature Representations for Music Annotation and Retrieval. In ISMIR (pp. 565-570).

Comment #5

In Section 4 It might make sense to include a section with different techniques that can be used mapping images to text. 

Comment #6

There is a number of machine learning techniques out there that might benefit in such an implementation. Thus, I would like to ask the authors to discuss the potential implementation of the proposed method to other machine learning techniques. Examples include hidden Markov models (HMM), regression, and graphical models. Here I suggest a few papers that use different machine learning techniques to solve such problems:

-- Abdel-Hamid, O., Mohamed, A. R., Jiang, H., & Penn, G. (2012, March). Applying convolutional neural networks concepts to hybrid NN-HMM model for speech recognition. In Acoustics, Speech and Signal Processing (ICASSP), 2012 IEEE International Conference on (pp. 4277-4280). IEEE. 

-- Mousas, C., Newbury, P., & Anagnostopoulos, C. N. (2014, May). Evaluating the covariance matrix constraints for data-driven statistical human motion reconstruction. In Proceedings of the 30th Spring Conference on Computer Graphics (pp. 99-106). ACM. 

-- Mousas, C. (2017). Full-body locomotion reconstruction of virtual characters using a single inertial measurement unit. Sensors, 17(11), 2589.

-- Chéron, G., Laptev, I., & Schmid, C. (2015). P-cnn: Pose-based cnn features for action recognition. In Proceedings of the IEEE international conference on computer vision (pp. 3218-3226). 

-- Zimo Li, Y. Z., Xiao, S., He, C., & Li, H. (2017). Auto-Conditioned LSTM Network for Extended Complex Human Motion Synthesis. arXiv preprint. arXiv, 1707. 

-- Saito, S., Wei, L., Hu, L., Nagano, K., & Li, H. (2017, July). Photorealistic facial texture inference using deep neural networks. In IEEE Conference on Computer Vision and Pattern Recognition, CVPR (Vol. 3). 

-- Bilmes, J. A., & Bartels, C. (2005). Graphical model architectures for speech recognition. IEEE signal processing magazine, 22(5), 89-100. 

-- Rekabdar, B., Mousas, C., & Gupta, B. (2019, January). Generative Adversarial Network with Policy Gradient for Text Summarization. In 2019 IEEE 13th International Conference on Semantic Computing (ICSC) (pp. 204-207). IEEE. 

-- Rekabdar, B., & Mousas, C. (2018, November). Dilated Convolutional Neural Network for Predicting Driver's Activity. In 2018 21st International Conference on Intelligent Transportation Systems (ITSC) (pp. 3245-3250). IEEE. 

-- Li, R., Si, D., Zeng, T., Ji, S., & He, J. (2016, December). Deep convolutional neural networks for detecting secondary structures in protein density maps from cryo-electron microscopy. In Bioinformatics and Biomedicine (BIBM), 2016 IEEE International Conference on (pp. 41-46). IEEE.

Given all my comments, I believe that this paper needs some improvement. I would be more than happy to reconsider my decision after a major revision.

Round 2

Reviewer 2 Report

I would like to congratulate the authors on their work. I sincerely believe that the effort made has substantially improved the understanding of the manuscript. Most of the recommendations made have been taken into account. In my opinion, it is a publishable work. However, I encourage the authors to continue improving the work by taking into account all the suggestions received.

Reviewer 3 Report

After carefully reading the revised version of the paper as well as the responses made by the authors of this paper, I feel confident that this is a strong and scientifically sound paper. For this reason, I would like to recommend this paper for the Sensors Journal. Well done!